# Raman micro-spectroscopy reveals the spatial distribution of fumarate in cells and tissues

Marlous Kamp [1,2,7], Jakub Surmacki [3], Marc Segarra Mondejar[4,5], Tim Young[4], Karolina Chrabaszcz[6], Fadwa Joud[2], Vincent Zecchini[4], Alyson Speed[4], Christian Frezza [4,5] & Sarah E. Bohndiek [1,2]

Aberrantly accumulated metabolites elicit intra- and inter-cellular pro-oncogenic cascades, yet current measurement methods require sample perturbation/disruption and lack spatio-temporal resolution, limiting our ability to fully characterize their function and distribution. Here, we show that Raman spectroscopy (RS) can directly detect fumarate in living cells in vivo and animal tissues ex vivo, and that RS can distinguish between Fumarate hydratase (Fh1)-deficient and Fh1-proficient cells based on fumarate concentration. Moreover, RS reveals the spatial compartmentalization of fumarate within cellular organelles in Fh1-deficient cells: consistent with disruptive methods, we observe the highest fumarate concentration ($37 \pm 19$ mM) in mitochondria, where the TCA cycle operates, followed by the cytoplasm ($24 \pm 13$ mM) and then the nucleus ($9 \pm 6$ mM). Finally, we apply RS to tissues from an inducible mouse model of FH loss in the kidney, demonstrating RS can classify FH status. These results suggest RS could be adopted as a valuable tool for small molecule metabolic imaging, enabling in situ non-destructive evaluation of fumarate compartmentalization.

Reprogramming of cellular metabolism is a key hallmark of cancer[1]. Perhaps the most prominent example of this is the existence of oncometabolites, aberrantly accumulated metabolites with pro-oncogenic capabilities[2]. Mutations in the gene encoding fumarate hydratase (FH) lead to profound cellular metabolic alterations and fumarate accumulation, which predispose to the rare hereditary leiomyomatosis and renal cell cancer (HLRCC) syndrome[3]. FH loss or transcriptional downregulation is also well established in a range of other cancers[4–7], implying a key role for fumarate in tumorigenesis.

The accumulation of fumarate leads to diverse biological consequences. For example, accumulated fumarate can react with cysteine residues of proteins, leading to the post-translational modification known as succination. Protein succination is a robust marker of FH loss in HLRCC tumours and it affects hundreds of proteins with important pathophysiological consequences, which are compartment specific[8]. In the mitochondria, fumarate-induced succination of the family of Fe-S cluster biogenesis proteins Iscu, Bola, and NFU1 decreases their availability for respiratory chain complexes, causing mitochondrial dysfunction[9]. In addition, mitochondrial fumarate can promote the vesicular release of mtDNA, causing the activation of innate immunity[10]. In the cytosol, fumarate-induced succination inactivates Keap1, leading to the subsequent stabilisation of the

[1]Department of Physics, University of Cambridge, JJ Thomson Avenue, Cambridge CB3 0HE, UK. [2]Cancer Research UK Cambridge Institute, Robinson Way, Cambridge CB2 0RE, UK. [3]Lodz University of Technology, Institute of Applied Radiation Chemistry, Laboratory of Laser Molecular Spectroscopy, Wroblewskiego 15, 93-590 Lodz, Poland. [4]Hutchison/MRC Cancer Unit, University of Cambridge, Biomedical Campus, Cambridge CB2 0XZ, UK. [5]CECAD, Joseph-Stelzmann-Straße 26, 50931 Cologne, Germany. [6]Institute of Nuclear Physics, Polish Academy of Sciences, Department of Experimental Physics of Complex Systems, Radzikowskiego 152, 31-342 Krakow, Poland. [7]Present address: Department of Chemistry, Utrecht University, 3584 CH Utrecht, The Netherlands. ✉e-mail: christian.frezza@uni-koeln.de; seb53@cam.ac.uk

antioxidant master gene NRF2[11], or it can form succinic glutathione, leading to oxidative stress and senescence[12]. Fumarate can also inhibit α-ketoglutarate (αKG)-dependent dioxygenases such as prolyl hydroxylases, leading to the aberrant stabilisation of HIF (Hypoxia-Inducible Factor) under normal oxygen levels. Nuclear accumulation of fumarate has a profound impact on cellular function, through inhibition of histone and DNA demethylases, leading to substantial epigenetic changes[13]. Fumarate is also implicated in genome stability and DNA repair; the ability of FH-deficient cells to respond to DNA damage is compromised, making them more sensitive to DNA damage[13]. Interestingly, the function of FH in DNA repair can be replaced by high concentrations of fumarate, partially compensating for the loss of FH[13].

Taken together, these lines of evidence indicate that fumarate exerts multiple biological functions in different cellular compartments, eliciting distinct cellular responses. To precisely understand how FH-deficient cells cope with the excess accumulated fumarate and unveil how diverse signals elicited by fumarate are coordinated, it is crucial to quantify how fumarate concentrations change over time and visualise its intracellular spatial distribution. Current methods to measure the intracellular concentration of fumarate rely on liquid chromatography-mass spectrometry (LC-MS), reporting average cellular concentrations of ~9 mM in FH-deficient cells[14]. Yet, LC-MS metabolite extraction protocols disrupt the intracellular architectures of a cell and require lengthy subcellular fractionations to interrogate concentrations in different compartments of the cell, hence are destructive methods that are not well-suited to studies in living cells or tissues. Furthermore, spatial information within these compartments is lost, with the readout providing an ensemble average across many cells or organelles. Small molecules such as fumarate are not amenable to staining for confocal fluorescence microscopy, since fluorescent labels are typically larger than the molecule under study and thus interfere with metabolism[15]. Recently, in vivo metabolic magnetic resonance imaging[16–18] has been used to track the production of [2,3-$^2$H$_2$]- or [1,4-$^{13}$C] malate in tumours after injection of labelled fumarate, however, this approach cannot probe the endogenous fumarate pool[19]. Knowledge of the spatiotemporal distribution of fumarate accumulation in, and trafficking between, cellular compartments thus remains elusive, making it difficult to understand how the many signals elicited by fumarate are coordinated.

Raman Spectroscopy (RS) is a label-free technique that distinguishes chemical compounds by the optical signature of their vibrational modes and has been widely applied in living cells[20–23]. In particular, Raman spectra from living cells are sensitive to changing concentrations of lipids[24–27], proteins[28], carbohydrates and nucleic acids[20,23], enabling identification and spatial localisation of the main cellular compartments[21,28]. Time-lapse studies can be applied to discriminate the emergence of oxidative stress[29], the internalisation and spatial distribution of therapeutic compounds[30,31], as well as the progress of the apoptotic cascade[32]. Nonetheless, while surface-enhanced Raman methods have shown promise in measuring metabolites in biofluids[33,34], spontaneous RS of small molecules in intact cells has thus far remained beyond the limit of detection[35].

Here, we demonstrate that RS can directly detect and map the oncometabolite fumarate in living cells. We first examine the fumarate Raman spectra in detail to understand the chemical origin of the observed vibrational modes. We then use Fh1-deficient and -proficient cell lines and mouse models to demonstrate the limits of detection and evaluate the performance of RS as a metabolic imaging tool. We show that RS can discriminate Fh1-deficient and -proficient cells and tissues, while revealing the previously unseen spatial distribution of fumarate concentrations according to the cellular compartment. Our findings open an avenue to metabolic imaging of small molecules, enabling the study of in situ metabolite distribution non-destructively in living cells and tissues.

## Results

### Theoretical and experimental Raman spectra of fumarate indicate a suitable limit of detection for application in cell studies

Density functional theory (DFT) simulations were performed to estimate the anticipated Raman cross sections of fumarate and comparable TCA cycle metabolites (succinate, malate, α-KG and oxaloacetate, Supplementary Fig. 1). While DFT simulations cannot provide a quantitative prediction of the spectral peak positions, due to the various ways in which a molecule can ionise, hydrogen-bond with water, or complex with available ions, they do give a general indication of relative cross sections and aid assignment of peaks to specific vibrational modes. All metabolites displayed prominent peaks in the 400–1800 cm$^{-1}$ (cellular "fingerprint" region) and 2800–3200 cm$^{-1}$ ("long wavenumber" region), covering carbon-carbon bond vibrations and C-H bond vibrations, respectively. Encouragingly, fumarate displayed up to 5-fold higher peak intensities in the fingerprint region, suggesting a lower limit-of-detection (LOD) compared to other TCA intermediates. DFT calculations examining a range of fumarate ions further show a general trend towards peaks at lower wavenumbers and with lower intensities during the transition from a non-hydrolysed state via monovalent ions to divalent ions of fumarate (Supplementary Fig. 2).

Next, we sought to compare the simulations to experimental Raman spectra acquired from powders and aqueous solutions. Comparing the sodium salts of primary TCA cycle intermediates (Fig. 1A, B), we see the highest Raman peak intensities for fumarate, indicative of a larger Raman cross section and in line with DFT simulations.

The fumarate powder spectrum is dominated by four peaks in the low wavenumber range at 913 ± 2 cm$^{-1}$, 1296 ± 2 cm$^{-1}$, 1431 ± 2 cm$^{-1}$ and 1657 ± 2 cm$^{-1}$. Spectral peak positions and intensities are expected to change in solution, as ionisation (depending on the cellular pH) and hydrogen bonding affect vibrational modes. We prepared aqueous solutions of metabolites using their disodium salts, which have a higher solubility than their acid counterparts. In these aqueous solutions, for many of the TCA cycle intermediates, the powder Raman peaks are absent or not distinguishable above the background (Fig. 1C). Again, fumarate stands out, displaying > 2-fold higher peak intensities in the fingerprint wavenumber range compared to metabolites. In solution, fumarate displays three dominant peaks at 1277 ± 2 cm$^{-1}$, 1401 ± 2 cm$^{-1}$ and 1652 ± 2 cm$^{-1}$ (Fig. 1D), indeed shifted to lower wavenumbers compared to the powder spectra, as predicted by DFT. The 1277 cm$^{-1}$ band is a C-H deformation ($\delta$) mode. The 1401 cm$^{-1}$ peak arises from a fully symmetric CO$_2^-$ stretch ($\nu$) vibration, and the 1652 cm$^{-1}$ band corresponds to the C=C stretch/CO$_2^-$ symmetric bending mode with a small shoulder peak belonging to C=C stretch/CO$_2^-$ asymmetric bending mode (see Supplementary Movies 1–4, c.f. refs. 36,37).

Finally, we examined a concentration dilution series to establish the sensitivity and limit-of-detection (LOD) for fumarate on our microscope. By integrating the area under the curve for fumarate bands, their Raman intensity (in cts cm$^{-1}$ s$^{-1}$) can be extracted as a function of fumarate concentration, serving as a calibration curve (Fig. 1E, F). A sensitivity of 19.2 ± 0.1 cts cm$^{-1}$ s$^{-1}$ mM$^{-1}$ was recorded for the 1401 cm$^{-1}$ peak for the current experimental setup at 26 mW incidence power. The LOD is defined here as the fumarate concentration above which the concentration series can no longer be fitted with a horizontal line; at a dwell time of 0.3 s, this occurs at 8 mM (Fig. 1F). Since signal-to-noise improves with longer dwell times, the LOD can be brought down to 4 mM for 5 s dwell time (Supplementary Fig. 3), however, longer dwell times are not suitable for undertaking area scans in live cells (see Methods). Although our LOD for 0.3 s dwell time is close to the reported LC-MS average concentration of fumarate in Fh1-deficient cells (9 mM)[14], local fumarate concentrations are expected to be higher in specific compartments, such as mitochondria, and accumulated to far higher levels in Fh1-deficient cells. Therefore, both

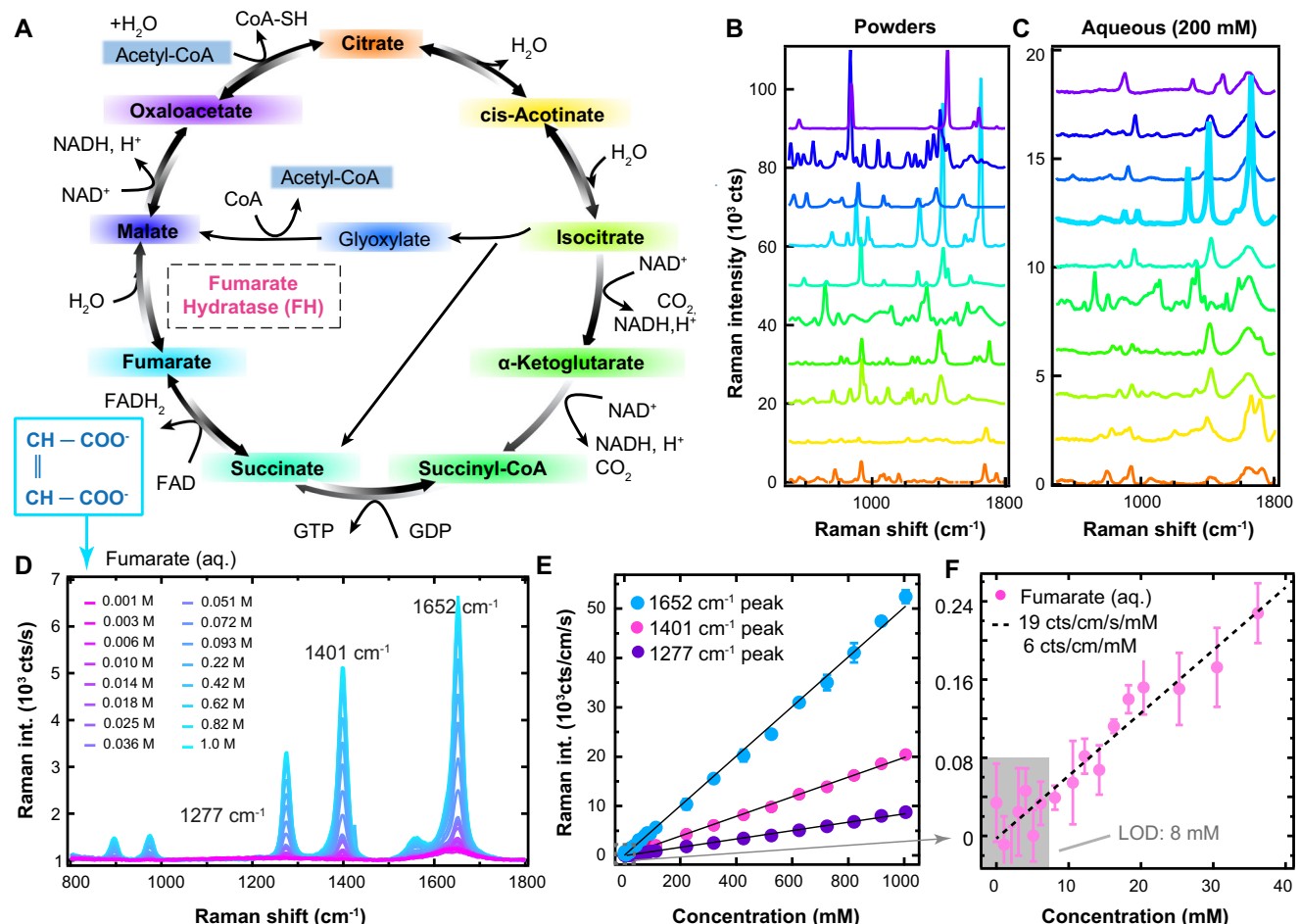

**Fig. 1 | Fumarate displays prominent Raman spectral features compared to other TCA cycle intermediates. A** Schematic displaying the TCA cycle. Each metabolite is labelled with a different colour and accompanied by its structural formula. The enzyme is labelled in pink. **B**, **C** Raman spectra of metabolites are displayed from (**B**) powders and (**C**) 200 mM aqueous solutions, taken with a 532 nm laser (26 mW power, 5 s resp. 2 s acquisition time, averages of 10 spectra). Spectra are vertically offset for clarity and colour coded according to the metabolites in (**A**); from bottom to top they also follow the metabolites in (**A**) clockwise from citrate. **D** Spectra of fumarate aqueous solutions at different concentrations (532 nm incident laser, 28 mW power, 5 s acquisition time). **E**, **F** Raman intensity (area under the curve, for 532 nm incident laser, 26 mW power, 0.3 s dwell time) as a function of fumarate concentration for (**E**) the three most prominent peaks of fumarate, and (**F**) for the 1401 cm⁻¹ peak close to the limit of detection (LOD). Data are represented as mean values ± SD of three separate measurements per concentration. $r^2$ values of the linear regressions are 0.99791 (1652 cm⁻¹), 0.998994 (1401 cm⁻¹) and 0.997838 (1281 cm⁻¹). Source data are provided as a Source Data file.

DFT calculations and experimental data from the pure compound indicate that RS has the potential to resolve fumarate in live cells.

## Fumarate can be resolved in Raman spectra of Fh1-KO cells

To examine the potential of RS to resolve fumarate accumulation, we used two previously characterised[2,38–40] mouse FH-deficient cell lines (*Fh1*$^{-/-\ CI1}$ and *Fh1*$^{-/-\ CI9}$) and their isogenic control (*Fh1*$^{fl/fl}$), which will be referred to as 'knock-out' (Fh1-KO) and 'wild-type' (Fh1-WT), respectively. Initially, we performed area scans of 20 individual cells per condition and obtained the average spectra across these scans (Fig. 2A). Raman spectra display four regimes: the 'fingerprint' region rich in Raman peaks of DNA, proteins and lipids (500–1800 cm⁻¹); a Raman silent region (1800–2750 cm⁻¹); the 'long-wavenumber' region of protein and lipids peaks (2800–3000 cm⁻¹); and two broad peaks caused by the symmetric and asymmetric stretch vibrations of water (>3000 cm⁻¹). We focus here on the fingerprint region, which is of greatest interest for the evaluation of fumarate concentrations based on our DFT and solution spectra results.

The fingerprint region is dominated by the CH₂ deformation mode[32] associated with lipids, as well as by the protein Amide I and III

bands and DNA bases (see annotations in Fig. 2A)[32,41–44]. Significant contributions also arise from phenylalanine and cytochrome C[30]. While the 1652 cm⁻¹ band of fumarate overlaps with the Amide I band and the bending mode of water, the 1401 cm⁻¹ mode is expediently located in the valley adjacent to the CH₂ deformation mode, apparently isolating it from other prominent peaks in the cell. Examining the 1401 cm⁻¹ mode, being the preferred band for fumarate detection, the average spectra of both Fh1-KO clones indeed displayed a higher Raman intensity than Fh1-WT cells (Fig. 2A inset) and the difference spectra confirm a significant contribution at 1401 cm⁻¹ (Supplementary Fig. 4A). The higher fumarate peak intensity persists when corrected for the generally higher Raman intensities of KO cells (Fig. 2A) after normalising by the area under the curve in the fingerprint region (Supplementary Fig. 4B). Using a PLS-DA model to classify the cells, 1280 cm⁻¹, 1401 cm⁻¹ and 1652 cm⁻¹ fumarate peaks feature prominently in the variable importance projection scores (Supplementary Fig. 5) demonstrating that accumulated fumarate is the main discriminator between the cell lines, with the model showing over 97% sensitivity and specificity for Fh1-WT and more than 93% sensitivity and specificity for the Fh1-KO clones (Supplementary Table 1). These

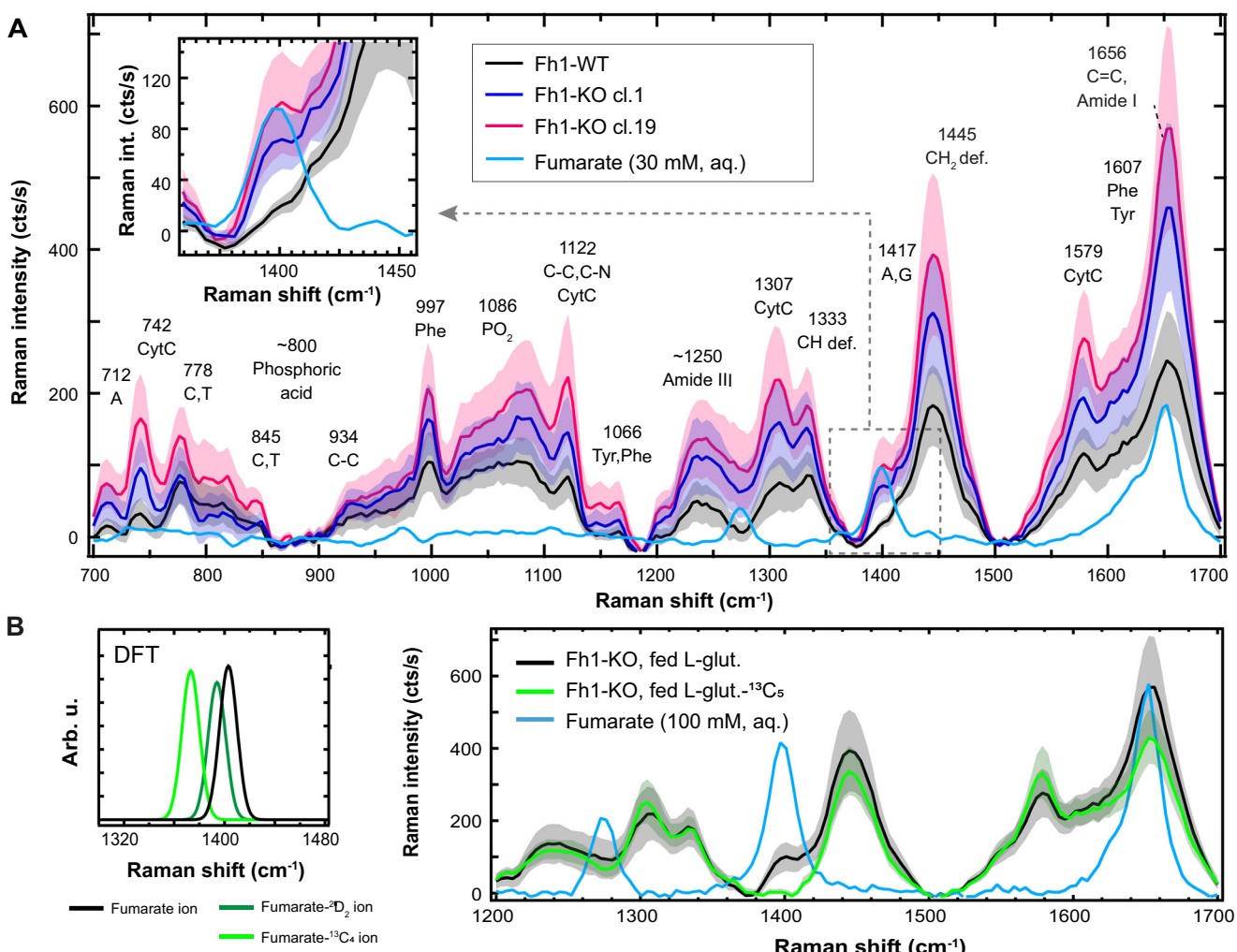

**Fig. 2 | Fh1-WT and Fh1-KO cell lines demonstrate distinct spectral properties associated with fumarate concentration. A** Averaged spectra of area scans (532 nm laser, 31 mW, 0.5 μm step size, 0.3 s integration time) for n = 20 individual cells for each of cell line in the fingerprint region. Black, purple and pink traces represent Fh1-WO, Fh1-KO clone 1 and Fh1-KO clone 19, respectively. The averaged spectrum for each cell was baseline-subtracted with a 10th order polynomial, after which all 20 scans were averaged. Graphs plus shaded areas represent mean values ± SD. The blue line displays a 30 mM disodium fumarate solution in water (532 nm laser, 26 mW, 5 s integration time), baseline subtracted with a 10th order polynomial and smoothed by a Savitzky–Golay filter (11 point window). Inset: Zoom at the wavenumber range around 1401 cm$^{-1}$. **B** Left: DFT calculations for fully dissociated ions of fumarate (black), deuterated fumarate (dark green) and fumarate-$^{13}C_4$ (light green), in the wavenumber range around 1401 cm$^{-1}$. Right: Average spectra for 10 Fh1-KO cells grown in DMEM supplemented with $^{13}C$-labelled L-glutamine (green) versus 20 Fh1-KO cells grown in DMEM with non-labelled L-glutamine (black). Graphs plus tinted areas represent mean values ± SD. For $^{13}C_5$-labelled L-glutamine, a full shift of the fumarate band was expected based on the DFT calculations. Peak assignments performed using refs. [43],[70]. Source data are provided as a Source Data file.

findings were also confirmed (using longer scan times) at 785 nm (Supplementary Figs. 6, 7), where the 1401 cm$^{-1}$ peak also becomes more pronounced due to the higher spectral resolution.

To further test whether the observed peak at 1401 cm$^{-1}$ stems from fumarate, Fh1-WT and Fh1-KO cells were cultivated in a medium with isotopically labelled glutamine–$^{13}C_5$, which generates fully labelled $^{13}C_4$-fumarate[38]. DFT calculations show that the 1401 cm$^{-1}$ peak would be expected to shift to 1373 cm$^{-1}$ for fumarate-$^{13}C_4$ (Fig. 2B left, Supplementary Fig. 8A). The Fh1-KO cells cultivated with L-glutamine-$^{13}C_5$ lacked the 1401 cm$^{-1}$ band, with a minor increase in the 1373 cm$^{-1}$ peak observed (Fig. 2B right, Supplementary Fig. 8B). No significant differences were observed in the peak positions of the Fh1-WT cells (Supplementary Fig. 8B). The relatively modest increase at 1373 cm$^{-1}$ may be due to the formation of a range of isotopically labelled fumarate species, each with their own characteristic vibrational modes. Taken together, these findings demonstrate the accumulation of fumarate upon Fh1 loss is resolvable using RS.

## Fumarate concentrations can be spatially resolved and vary significantly between cellular compartments

Comparing cell spectra (Fig. 2A) with fumarate calibration curves (Fig. 1E, F) enables the extraction of cellular fumarate concentrations (Fig. 3A). In WT cells, the bulk of fumarate appears to lie below the LOD, hence cannot be accurately determined. Average fumarate concentrations in Fh1-KO cells derived from these histograms are 10 ± 2 mM (mean ± standard error among cells for Fh1-KO cl.1) and 20 ± 4 mM (for Fh1-KO cl.19), which are in the range previously reported. Maximum apparent concentrations reach up to 60 mM in Fh1-KO cl.19.

K-means clustering was applied to all area scans per cell line to partition the spectra by cellular compartment. The k-means cluster maps and their associated loadings (here called set *K1*) are mostly determined by the long-wavenumber bands, which are most intense (Fig. 3B, C). The cluster maps visualise the spatial organisation of the cell into nuclear, cytoplasmic, mitochondrial, and membrane regions,

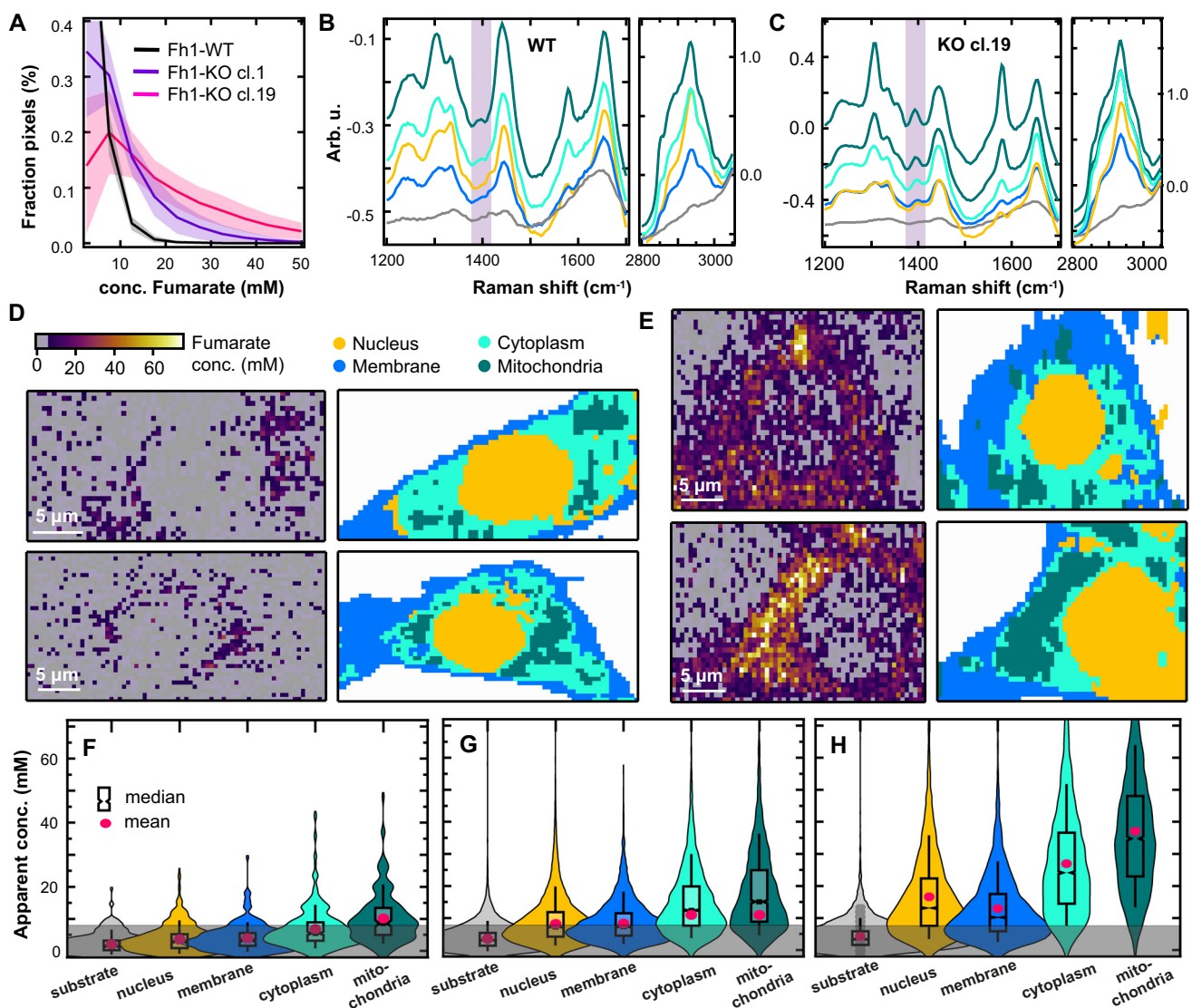

**Fig. 3 | Raman spectroscopic imaging reveals spatial compartmentalization of fumarate. A** Fumarate concentrations in Fh1-WT cells ($n = 20$, black) and Fh1-KO cells ($n = 20$, purple: cl.1, pink: cl.19), normalised by the number of pixels per area scan (bin width: 5 mM). Histograms are presented as mean values ± SD. **B**, **C** K-means clustering loadings extracted from the combined analysis of (**B**) 20 Fh1-WT cells and (**C**) 20 Fh1-KO cl.19 cells. Loadings are assigned to substrate (grey), nucleus (yellow), membrane (blue), and mitochondria (green). For Fh1-KO cells, two classes are assigned to the mitochondria. **D** Exemplar fumarate concentration maps (left) and k-means clustering maps (right) for two Fh1-WT cells. The labelling of the k-means cluster maps corresponds in colour to the loadings in (**B**, **C**). **E** As (**D**) but for two Fh1-KO cl.19 cells. **F**–**H** Fumarate concentrations plotted per cluster class for (**F**) the 20 Fh1-WT cells from (**A**, **B**), (**G**) the 20 Fh1-KO cells cl. 1 cells from (**A**), and (**H**) the 20 Fh1-KO cells cl. 19 cells from (A,C). Values are tabulated in Table 1. The box plot indicates the median value with 95% confidence interval (sloped edges), 1st/3rd quartile (box edges), and 9th/91st percentile whiskers. The pink dot indicates the mean fumarate concentration in the cluster class. The grey-shaded area marks concentrations below the estimated LOD. Source data are provided as a Source Data file.

as confirmed by performing a peak assignment. Primarily, the k-means loadings of the mitochondria are characterised by prominent peaks of cytochrome C (Supplementary Fig. 9), which is indeed located in the mitochondrial membrane[30,31,45,46]. For WT cells, a faint band at 1401 cm⁻¹ is present throughout and a second faint band at 1388 cm⁻¹ becomes more pronounced from nucleus to mitochondria. These faint bands can be attributed to the oxidised and reduced states of cytochrome C, as previously observed in mitochondria via SERS[46] (see also the shoulder peak in Supplementary Fig. 9). Thus, the 1401 cm⁻¹ band of fumarate is not completely isolated from underlying cellular bands as anticipated from the average cell spectra. A further potential confounder could be the complexes formed by excess fumarate with cysteine and the cysteine-containing protein glutathione, which could alter the fumarate vibrational modes. S-(2-succinyl)glutathione was observed to display a band at 1413 cm⁻¹ of negligible intensity

compared to fumarate itself even for a >100 mM solution (Supplementary Fig. 10). Therefore, we deem it unlikely this fumarate adduct is contributing to signals in the cell, but it is possible that such complex formation could reduce observed fumarate concentrations measured by the 1401 cm⁻¹ band. Nonetheless, the fumarate band at 1401 cm⁻¹ is intense and becomes increasingly visible in the loadings from nucleus to mitochondria for Fh1-KO cells, giving a first insight into its distribution.

Having confirmed the k-means cluster assignments using a targeted mitochondrial Raman probe (see Supplementary Note 1 and Supplementary Fig. 11), we then mapped the spatial fumarate concentrations in the cell. In a qualitative comparison of the fumarate locations (Fig. 3D, E, left) and k-means maps for KO cells (Fig. 3D, E, right), fumarate is located preferentially outside the nucleus. By comparing k-means cluster maps and fumarate concentration maps,

the average fumarate concentration in each identified organelle was calculated (Table 1). Although average fumarate concentrations differ slightly per cell, they tend to increase from relatively lower levels in the nucleus and cell membrane to the cytoplasm and still further in the mitochondria (Fig. 3F–H). Similar concentrations were found when extracted by fitting a Gaussian peak shape to the 1401 cm⁻¹ band, which reduces the influence of parameter choices as it does not require baseline fitting, but is more susceptible to noise (Supplementary Fig. 12). Some fumarate signal appears outside of the cellular region, which arise from the fitting procedure being performed on the entire imaging region, rather than masked only to the cellular region; it is still possible for the fitting procedure to converge when applied to background spectra in some instances, giving an impression of the background noise in the imaging data. Fumarate concentrations in the cytoplasm and mitochondria are statistically significantly different from the membrane in a one-way ANOVA test for KO cl.19 cells (p = 10⁻²⁶).

Although Fh1-WT cells have intact FH activity, apparent fumarate concentrations surpass the LOD in the mitochondria, which may be due to the weak contribution of cytochrome C related vibrational modes in a similar wavenumber region. Removing this background contribution measured in the FH-proficient cells, we estimate true average fumarate concentrations of Fh1-KO cl.19 to be 5 mM in the

nucleus, 10 mM in the cell membrane, 18 mM in the cytoplasm and 27 mM in the mitochondria (Table 1).

## RS reveals fumarate in excised tissues from an inducible mouse model of FH-loss in the kidney

To assess whether fumarate can also be detected in tissues, we examined tissues from an inducible transgenic mouse model of Fh1-loss[10] (two *Fh1*$^{fl/fl}$, two with induced FH loss: *Fh1*$^{-/-}$) using RS in a blind identification test. Raman images show a clear distinction between fumarate intensities in different sections (Fig. 4A), with fumarate concentrations up to 40 mM detected in tissues that were subsequently unblinded as *Fh1*$^{-/-}$. Variations in concentration are visible over each section, which may point to a distinct localisation of fumarate within different regions of the kidney. *Fh1*$^{-/-}$ tissues exhibit fumarate concentrations largely above the LOD, successfully enabling the identification of each genotype (Fig. 4B).

## Discussion

We show here the potential of RS to detect and spatially resolve fumarate in situ. We identify the Raman fingerprint of the fumarate molecule and show a linear concentration dependence down to an LOD of ~8 mM. We found that the distinctive Raman peak arising from the symmetric $CO_2^-$ stretch vibration (1401 cm⁻¹) could be distinguished in live murine kidney Fh1-KO cells but was not apparent in Fh1-WT cells. Moreover, the peak height varied by cellular compartment, identified with k-means clustering, allowing the quantification of fumarate in a compartment-specific manner. The average concentrations detected show good agreement with previously obtained average fumarate concentrations from LC-MS and recent confirmation of increased cytosolic fumarate[47]. When the clusters were mapped spatially, fumarate concentration showed minimal distribution in Fh1-WT compared to Fh1-KO cells. Finally, when RS was applied blindly to kidney tissue sections from a transgenic mouse model, the 1401 cm⁻¹ peak was diagnostic of loss of Fh1 and showed similar concentrations as found in vitro.

Being able to accurately quantify and establish the intracellular distribution of fumarate is crucial for understanding the activation of the many molecular cascades orchestrated by this metabolite. For instance, the ability to image fumarate at the single cell level with this method will help to appreciate the intracellular heterogeneity and

**Table 1 | Apparent average (mean ± standard deviation) fumarate concentrations for each cluster class, per cell line**

| Apparent average fumarate concentrations | | | |
|---|---|---|---|
| | Fh1-WT conc. (mM) | Fh1-KO Cl. 1 conc. (mM) | Fh1-KO Cl. 19 conc. (mM) |
| Substrate | 2 ± 3 | 4 ± 4 | 3 ± 3 |
| Nucleus | 4 ± 4 | 8 ± 7 | 9 ± 6 |
| Cell membrane | 4 ± 4 | 8 ± 6 | 14 ± 10 |
| Cytoplasm | 6 ± 6 | 11 ± 8 | 24 ± 13 |
| Mitochondria | 10 ± 8 | 11 ± 9 | 37 ± 19 |

The standard deviation represents the variability of all measured concentrations in each cluster class.

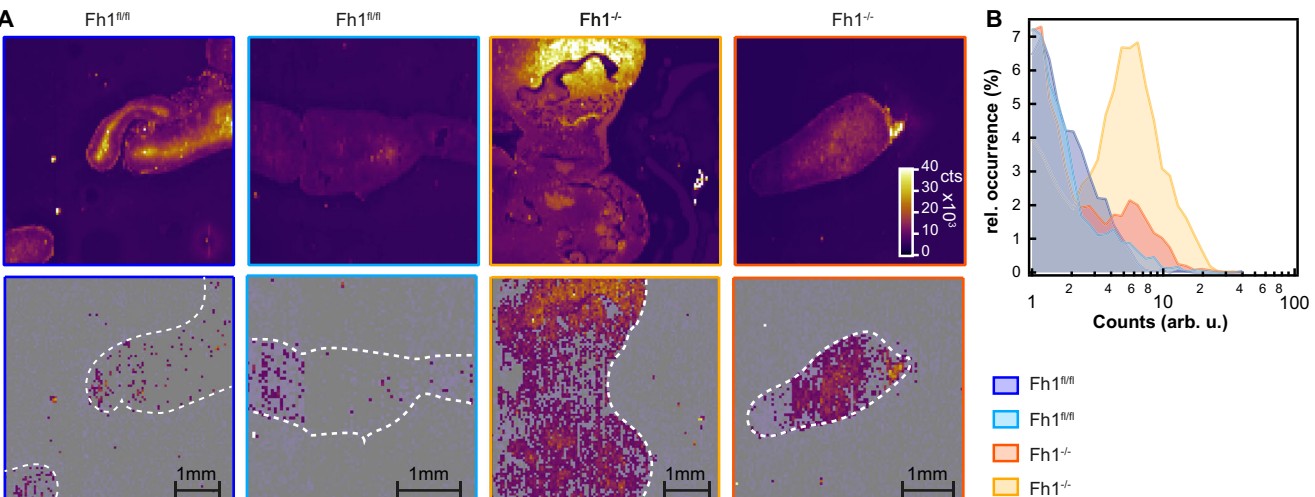

**Fig. 4 | Fumarate mapping in tissue section. A** Top: Raman intensity maps for the 1130 cm⁻¹ band, used to show the outline of the mouse kidney tissue section. Bottom: Fumarate concentration maps obtained by fitting the 1401 cm⁻¹ band. The white outline traced from the Raman intensity maps serves as a guide to the eye. **B** Relative occurrence of fumarate concentrations in each area scan, normalised by scan size. Measured concentrations likely vary from true concentrations due to variations in focussing over the specimen, however, *Fh1*$^{fl/fl}$ tissues can be distinguished from induced tissues *Fh1*$^{-/-}$ by whether the majority of calculated concentrations are above the LOD. Source data are provided as a Source Data file.

clonality of FH-deficient cells or tumour tissues. Application to tissues enables the investigation of how cancer cells interact with components of the tumour microenvironment and whether fumarate exchange occurs. It would also be possible with this method to visualise the time course of increasing fumarate accumulation in different compartments of a cell longitudinally following modulation of FH expression, to improve understanding of how this elicits a cellular response. Moreover, we show through DFT calculations and experimental studies in solutions evidence of the potential of RS to be used beyond application in fumarate biology, with promise shown for detecting other metabolites in the TCA cycle.

RS is a well-established analytical technique underpinned by the availability of robust commercial instruments, which makes the use of RS for this purpose immediately accessible to the community. Despite our promising findings, the LOD appears insufficient to detect the full range of fumarate concentrations observed in Fh1-WT cells, restricting the application of spontaneous RS to the study of pathophysiological states. The low sensitivity arises because the Raman scattering process has an extremely low probability, which we compensated by long exposure times (~10 min per cell) in our spontaneous Raman microscope. Future application to longitudinal studies or detection of other metabolites will, however, likely require an increase in sensitivity of the technique due to their lower Raman scattering cross sections.

Future studies that require lower LODs or faster scanning could adopt coherent Raman techniques such as stimulated (photothermal) Raman spectroscopy (SRS)[48], coherent anti-Stokes Raman spectroscopy (CARS)[49] or surface-enhanced Raman spectroscopy (SERS)[50–53], which provide resonant signal-enhancements and fluorescence-background suppression[54]. Although coherent Raman techniques require more specialist microscopy equipment, they also afford substantially greater sensitivity and imaging speed compared to conventional spontaneous Raman microscopy, offering a promising route to enable broader metabolite imaging and quantification using the Raman scattering process.

Additionally, while the 1401 cm$^{-1}$ peak can be distinguished, it sits on the shoulder of an adjacent cellular peak at 1445 cm$^{-1}$, associated with $CH_2$ mode in DNA, making quantification challenging. We also identified other potential confounders at that peak location, with weak contributions possible from cytochrome C and S-(2-succinyl)glutathione, which likely underpin at least some of our measured signal from Fh1-WT cells. Stable isotope probing techniques are likely to add value in this context, which we already examined by the application of L-glutamine-$^{13}C_5$, which removed the characteristic fumarate peak at 1401 cm$^{-1}$. An alternative approach could be to use deuterium labelling, since deuterium peaks typically appear in the silent region of the cellular Raman spectrum, thus improving signal-to-noise ratio by locating the identifying peak away from the main cellular vibrational modes. The application of stable isotope probing should be examined in more detail in future work, in combination with coherent Raman microscopy techniques, to maximise the sensitivity of the method and explore the potential for dynamic metabolic analysis.

Finally, fumarate accumulation induces a host of cellular responses that could also be detected by RS, for example, oxidative stress[29,55–57], which we did not address in this study. A potential further benefit of applying RS in metabolite studies is the capability to resolve other cellular responses from changes in the cellular vibrational spectra, which are captured simultaneously with the metabolic information. To tease apart these different contributions to the Raman spectra, machine learning methods that have already shown promise in a range of RS applications could be applied[58]. For example, non-negative matrix factorisation and principal component analysis have assisted in analysing radiation response biomarkers and hypoxia indicators in cells and tissues[59], while a convolutional neural network has been applied for metabolomics on liver carcinoma tissue[60].

Adopting machine learning may also aid in the detection of other oncometabolites with less expediently located vibrational modes, such as succinate.

Overall, this study introduces RS as a tool to enable in situ mapping of fumarate, opening avenues for the study of fumarate accumulation and spatial compartmentalization in cells and within tissues in pathophysiological processes.

## Methods

All animal experiments were conducted in accordance with all relevant ethical regulations under the United Kingdom Animals (Scientific Procedures) Act, 1986, approved through project licence number P8A516814.

### DFT calculations

Density functional theory (DFT) calculations were performed on major TCA cycle metabolites (including different hydrolysis states of fumarate, succinate, malate, α-ketoglutarate and oxaloacetate) using the hybrid B3LYP exchange-correlation energy functional[61–64]. Starting structures were generated using Chem3D 16.0 software (CambridgeSoft). To account for the solvent, a self-consistent reaction field was applied, more specifically the polarisable continuum model for water. Geometry optimisations were performed using the Def2GFP basis set, except for disodium fumarate and oxaloacetate where the Popletype 6-311 G* basis set was used. Raman frequency calculations were then performed on the optimised structures using the Gaussian '09 ab initio program package[65].

### Chemical compounds

Powder metabolites were obtained from Sigma Aldrich: cis-aconitic acid (≥98%, CAS 585-84-2, cat. no. A3412), coenzyme A sodium salt hydrate (CAS 55672-92-9, cat. no. C3144), sodium fumarate dibasic (≥99%, CAS 17013-01-3, cat. no. F1506), sodium glyoxylate monohydrate (≥ 93%, CAS 918149-31-2, cat. no. G4502), oxalic acid disodium salt (≥99.99% trace metals basis, CAS 62-76-0, cat. no. 379735), sodium succinate dibasic (≥ 98%, CAS 150-90-3, cat. no. 14160), and L-glutamine-$^{13}C_5$ (98%, CAS 184161-19-1, cat. no. 605166).; from Acros Organics: DL-Isocitric acid trisodium salt hydrate (95%, CAS 1637-73-6, cat. no. 10388050), and citric acid monohydrate (for HPLC, CAS 5949-29-1, cat. no. 16514274); from Insight Biotechnology Ltd: DL-Malic acid disodium salt (CAS 676-46-0, cat. no. SC-234826); and from Scientific Laboratory Supplies Ltd: α-ketoglutaric acid disodium salt dihydrate (>=98.0%, CAS 305-72-6, cat. no. 75892). Cytochrome C (horse heart muscle, 90%, CAS 9007-43-6, cat. no. 147531000) was obtained from Acros. The following isotopically labelled compounds were used: L-glutamine-[2,3,3,4,4-$^2H_5$, 97 atom%] (98% purity, CAS 14341-78-7, cat. no. DLM−1826), L-glutamine−$^{13}C_5$ (99% purity, CAS 184161-19-1, cat. no. CLM-1822-H-PK) and succinic acid disodium salt-[2,2,3,3-$^2H_4$, 80 atom %] (95% purity, CAS 203633-14−1, cat. no. DLM-2307) from CK Isotopes Ltd.; and fumaric acid−1,4−$^{13}C_2$,2,3-$^2H_2$ (≥99 atom% 13C, ≥98 atom% $^2H$, CAS 100858-52-4, cat. no. 752576) from Sigma Aldrich. All chemicals were used as received.

### Preparation of chemical compounds for RS

To measure metabolites in solution, powders were dissolved in deionized water (18.2 mΩ cm) to a concentration of 200 mM to remain below saturation. Concentration dilution series of fumarate and succinate were prepared as follows. First, powders were dissolved to a 1 M stock solution (total volume: 100 mL), after which dilutions in steps of 100 mM were made by combining stock solution with deionized water to a total volume of 10 mL in polypropylene centrifuge tubes (Fisherbrand). A total volume of 100 mL was prepared of the 100 mM solution, which was similarly used as a stock solution to prepare lower concentrations in steps of 10, 5 or 2 mM.

## Monolayer cell culture and cell preparation for RS

We used Fh1-proficient and -deficient cell lines, both of which are adherent epithelial murine kidney cells provided by Prof. Christian Frezza (CF) and originally derived as described in ref. 38. Briefly, in this method, immortalised Fh1-proficient epithelial kidney cells from Fh1 conditional knock out mice are infected with recombinant adenovirus expressing Cre recombinase to target the Fh1 alleles. We cultured two genotypes: the wildtype Fh1-proficient line (*Fh1*$^{fl/fl}$, hereafter denoted Fh1-WT) and knock-out Fh1-deficient line (*Fh1*$^{-/-}$, hereafter denoted Fh1-KO). Two clones of the Fh1-KO cell line were used, denoted cl.1 and cl.19. Cell lines were tested for mycoplasma contamination before use.

Cells were cultured in Dulbecco's Modified Eagle Medium (DMEM 1X, containing pyruvate and 4.5 g/L D-glucose, L-glutamine from Gibco, Life Technologies Billings, MT, USA, cat. no. 41966-029) supplemented with 10% heat-inactivated foetal bovine serum (FBS, Gibco, cat. no. 10500-064) and 5% penicillin streptomycin (PenStrep, Gibco, cat. no. 15070-063). Cells were cultured as monolayers in T75 polystyrene cell culture flasks in a 37 °C humidified incubator with 5% $CO_2$. Cell suspensions were obtained by flushing away dead or detached cells with a phosphate buffer saline (pH 7.4, Gibco) and treating with 0.25% trypsin-EDTA (1X, Gibco) at 37 °C for 4 min. Cell suspensions were then centrifuged at 1000 $g$ for 4 min, forming a pellet, which was resuspended in DMEM. For regrowth, an appropriate number of cells was placed in a T75 flask containing 12 mL DMEM, with typical cell densities of $3 \times 10^3 - 2 \times 10^4$ cells/mL for wildtype cells and $2 \times 10^4 - 3 \times 10^4$ cells/mL for knock-out cells. Additional uridine (50 μg/L, ≥99%, CAS 58-96-8, cat. no. U3750, Sigma Aldrich) was supplied to the medium.

For RS, cells were seeded onto quartz cover slips (25 mm diameter, thickness no. 1.5, UQG Optics, Cambridge, UK) placed into 6 well plates (Corning Ltd., Deeside, UK) with 2 mL of cell suspension on top. Cells were imaged 2 to 4 days after seeding, at sub-confluency. To remove fluorescence from the medium before RS, quartz plates were gently rinsed with a medium of phenol red-free DMEM (Gibco, cat. no. 31053-028) supplemented with FBS, PenStrep, 1.2 mM pyruvate (100X, Gibco, cat. no. 11360-039) and 5 g/L (34 mM) L-glutamine (Gibco, 200 mM, cat. no. 25030-024) – hereafter called "imaging medium". The cover slips were then mounted into an Attofluor™ cell chamber (Invitrogen, Waltham, MD, USA, cat. no. A-7816) with imaging medium supplied on top.

For mitochondrial labelling, compound **1**[66], referred to hereafter as mitokyne, was prepared and provided by Prof. Duncan Graham at the University of Strathclyde. The compound was dissolved in DMSO to a concentration of 1 mM. 10 μL of mitokyne solution was added to a well containing a quartz cover slip with cells of interest and 2 mL medium, followed by incubation for 30 mins. The cover slip was rinsed twice with imaging medium before placing in the cell chamber.

## Animal handling and tissue harvesting

Mice were of mixed genetic background C57BL/6 and 129/SvJ. Animals were bred and maintained under specific pathogen free conditions at the Breeding Unit (BRU) at the CRUK Cambridge Institute (Cambridge, UK). The tissues used in the present study were derived from the cohort of an inducible transgenic mouse model previously reported by co-authors Zecchini and Frezza[10] and based on crossing *Fh1*$^{fl/fl}$ and *R26Creert2* mice, which were gifts from Prof Gottlieb (RBNI, Technio, Israel Institute of Technology)[67] and Dr Winton (CRUK, Cambridge Institute, Cambridge, UK)[68], respectively. Experimental mice were homozygous for the conditional *LoxP-exon3/4-LoxP Fh1* allele and expressed the Cre–recombinase-Ert2 fusion under control of the *Rosa*26 promoter (*Fh1*$^{fl/fl}$; *R26 Cre-Ert2/Cre-Ert2*). Littermate controls lacked the *LoxP-exon3/4-LoxP* allele but also expressed the Cre-Ert2 allele under the control of the *Rosa26* promoter (*Fh1*$^{+/+}$; *R26 Creert2/ Creert2*). Control mice were induced and sacrificed at the same time as their experimental littermates. Extensive validation of tissue Fh1 expression, along with fumarate and 2-succinocysteine levels, were presented in the prior study[10].

For tamoxifen preparation and treatment, 5 mL ethanol were added slowly to 1 g tamoxifen (Sigma-Aldrich) in a 50 mL Falcon tube. The tamoxifen/ethanol mix was sonicated at 40% amplitude in 20 s pulses until the Tamoxifen was completely dissolved. 50 mL of corn oil (Sigma-Aldrich) pre-heated at 50 °C were added immediately to the tamoxifen/ethanol mix to obtain a 20 mg/mL Tamoxifen stock solution. The tube was then vortexed and incubated for up to 12 h in an orbital shaker at 50 °C to ensure adequate solubilisation. Aliquots of 5 mL were then stored indefinitely at −20 °C. Prior to injection, the mixture was allowed to warm to room temperature. Age-matched mice (between 10 and 12 weeks old) were used in all experiments. Each animal received three doses of 2 mg tamoxifen each via intraperitoneal injection. Tamoxifen was administered at ten-day intervals to allow animals to recover between doses. Twenty-five days after the first injection, animals were killed by neck dislocation and the kidneys were speedily collected and snap frozen in liquid nitrogen. Tissues were frozen-sectioned onto Thermo Scientific SuperFrost Plus slides, with a thickness of 6 μm and a minimum separation of 300 μm between sections.

## Confocal Raman micro-spectroscopy

RS was performed on an inverted confocal Raman microscope (Alpha 300 M+, WITec GmbH, Germany) equipped with a 532 nm laser (WITec GmbH, Germany) and a 785 nm single mode diode laser (XTRA II; Toptica Photonics Inc., USA). The output signal was coupled into a 300 mm triple grating imaging spectrometer (Acton SpectraPro SP-2300; Princeton Instruments Inc., USA) with a 600 l/mm grating (500 nm blaze) and a 1200 l/mm grating (750 nm blaze), coupled into a 100 μm single-mode fibre. Spectra were recorded on a thermoelectrically cooled CCD camera (DU401A-BV; Andor, Ireland) using Control FOUR of the WITec Suite software (WITec GmbH, Germany).

The spectral resolution of the Raman spectra was 3.61 cm$^{-1}$ (532 nm laser, 600 l/mm grating) and 1.65 cm$^{-1}$ (785 nm laser, 1200 l/mm grating). A 60x water immersion objective with cover glass correction collar (Nikon Plan Apo IR 60x, NA 1.27) was placed against the cover slip for imaging. For this objective, a 532 nm laser has a theoretical lateral resolution of 214 nm and spot size of 511 nm. The spatial resolution was determined by scanning over the edge of a silicon wafer. A silicon wafer can be split cleanly along a crystal plane, to create an edge response. Differentiating the edge response gave the line spread function (LSF), and the spatial resolution of the microscope was taken to be the full width at half maximum (FWHM) of the LSF. A spatial resolution $(0.35 \pm 0.04) \cdot 10^3$ nm at 532 nm incidence and $(1.5 \pm 0.2) \cdot 10^3$ nm at 785 nm incidence (Supplementary Fig. 13) were confirmed. The wavenumber scale was calibrated in the fingerprint region using known peak positions of biphenyl-4-thiol (powder, 97%, CAS 19813-90-2, Aldrich). Before measuring each sample, a silicon wafer was measured to ensure reproducible intensities, ensuring $(8.0 \pm 0.3) \cdot 10^2$ counts mW$^{-1}$ s$^{-1}$ for the primary band.

To measure dilution series of fumarate, 1 mL solution was placed onto a quartz cover slip in a cell chamber. The objective was focused into the solution just above the quartz cover slip, mimicking cell measurements. For spectra of dry metabolites and 200 mM solutions, a few mg of powder or 1 mL of solution was placed in a well of a μ-Slide 8 Well plate (Ibidi, with #1.5 polymer cover slip). Spectra were acquired by focusing beyond the cover slip of the 8-well plate onto the powder or into the solution, moving the objective until the polymer cover slip's fingerprint spectrum had disappeared.

To enable live cell imaging, the microscope was equipped with a custom chamber to control the temperature (37 °C) and $CO_2$ level (5%) (Digital Pixel). Cells were seeded onto unmodified fused quartz cover slips, which only display intrinsic Raman signals only below 500 cm$^{-1}$, outside of the wavenumber range of interest for live cells[69].

Additionally, these cover slips are thin enough (0.17 mm, #1.5) to use with a short working distance water immersion objective, in contrast to widely used calcium fluoride windows.

For tissue imaging, sections were imaged through a quartz cover slip placed directly on top of the defrosted tissue. Raman spectroscopy measurements were performed blinded on sections of these kidney tissues and tissues were assigned to *Fh1*^fl/fl or *Fh1*^−/− groups based solely on the Raman data; their experimental status was confirmed only after these assignments were performed.

Data acquisition for spontaneous RS requires a careful trade-off between resolution, step size, integration time, and total imaging time. To determine the optimal conditions for mapping, we compared 532 nm and 785 nm laser wavelengths available on our instrument. Similar laser wavelengths, laser power and integration times have been used successfully in literature to map cells[30,70,71]. Raman cross sections scale to the fourth power with laser frequency[32]. Integration times of ~30 s were required to achieve sufficient signal-to-noise ratio for peak fitting at 785 nm excitation at 90 mW power (on sample, as determined with an optical power metre, Thorlabs PM100D), which led to a scan time of ~120 min to cover the area of a single cell. At 532 nm excitation at 26 mW power, an integration of 0.3 s was found to provide comparable signal-to-noise ratio, enabling single cell coverage in ~10 min with a spatial step size of 0.5 μm, albeit with a slightly poorer spectral resolution compared to 785 nm due to the grating characteristics. The shorter imaging duration is highly advantageous, being less susceptible to cellular motion and enabling a higher throughput of data acquisition. Nonetheless, illumination at 532 nm has a higher potential to cause long-term damage to the cells with extended illumination at ≥5 mW[42,72]. In our study, cell morphology was unchanged during <15 min scanning time and no change in chemical composition was observed. To confirm the expected damage threshold in the cell line under study, we illuminated the same spot on one cell repeatedly at 0.5 s acquisition time with 0.5 s pause, observing spectral changes only after 200 s, suggesting that our experimental parameters are below the threshold at which biological changes would be expected (Supplementary Fig. 14). For PLS-DA, data consisted of line scans taken through a major axis of the cell in 20 steps at 5 s acquisition time (532 nm incidence, 26 mW power) or 10 steps at 30 s acquisition time (785 nm incidence, 120 mW power), to maximise signal-to-noise ratio while still representing the spectral contributions from the major cellular compartments.

### Data processing

Data processing was performed in Igor Pro (version 8.0.4.2). Cosmic ray removal was applied to all spectra before analysis. To analyse fumarate concentration dilution series, a flat baseline was subtracted (the average number of counts between 1324–1379 cm⁻¹, the wavelength range preceding the fumarate band of interest) and the area under the curve (between 1399 - 1419 cm⁻¹) was integrated, leading to a calibration curve as a function of fumarate concentration. For baseline subtractions in cellular spectra, we used an iterative algorithm fitting a polynomial (Supplementary Fig. 15). To visualise differences between the fingerprint areas of different cell lines, map-scans of each cell were averaged and then baseline subtracted with a tenth order polynomial (10 iterations). To extract fumarate concentrations from individual cellular spectra, we applied a 4$^{th}$ order polynomial baseline subtraction (7 iterations with a 'tolerance' of 15 counts, which was taken as twice the experimental noise floor, Supplementary Fig. 16), and integrated the area under the baseline-corrected curve. No Savitzky-Golay filter[73] was applied before processing, since such smoothing was found to affect extracted concentrations depending on the chosen smoothing window. Alternatively, a Gaussian peak shape was fitted to the spectrum without baseline subtraction, and the fitted peak height was used as a measure for fumarate concentration.

Fumarate concentration dilution series were analysed using linear regression, with $r^2$ values for the goodness of fit reported. Histograms of fumarate concentration distribution according to spatial compartments were constructed using 5 mM bins.

Partial least squares discriminant analysis (PLS-DA) was performed to test the ability of RS to discriminate between the Fh1-WT and Fh1-KO clones using the PLS-Toolbox (MATLAB, Mathworks). The 532 nm spectra were split into sets for calibration [n(WT) = 600, n(KO cl. 1) = 615, n(KO cl. 19) = 585] and validation [n(WT) = 200, n(KO cl. 1) = 205, n(KO cl. 19) = 195] by removing every fourth spectrum to form the validation set. The 785 nm spectra were split into sets for calibration [n(WT) = 450, n(KO cl. 1) = 450, n(KO cl. 19) = 450] and validation [n(WT) = 150, n(KO cl. 1) = 150, n(KO cl. 19) = 150] similarly. Spectra were normalised to the area under curve. Cross validation was performed using venetian blinds and 10 data splits, and the model was built using 7 latent variables.

Clustering algorithms are routinely used in Raman spectroscopy to spatially delineate compartments of the cell[24,28,30,70,71,74], since different organelles contain characteristic ratios of proteins, lipids, and RNA/DNA[75]. K-means clustering was performed here using random initial assignment of spectra to a class, and by evaluating distances as Manhattan distances. K-means clustering was applied separately to 20 area scans per cell line (20 Fh1-WT cells, 20 Fh1-KO cl1, and 20 Fh-KO cl.19), totalling 41647, 41671 and 43027 spectra, respectively. As a pre-processing step to reduce the effect of possible illumination differences among scans, spectra were mean-centered, i.e. each spectrum was divided by the average cell spectrum and then multiplied by its standard deviation. The total number of classes was tailored per cell line, since classes assigned to the substrate varied between 2 and 4. The number of initial classes chosen (8 for WT cells and for KO cl.1 cells, 6 for KO cl.19 cells) was such that 4–5 classes remained for the cellular spectra corresponding to major organelles.

### Statistics and reproducibility

Statistics during processing were performed as described in the previous section. The number of 20 cells per cell line was chosen based on consideration of the time needed to image a single cell. No statistical method was used to predetermine the sample size. No data were excluded from the analyses. The investigators were blinded to allocation during Raman imaging of tissues, but not during Raman imaging of cell lines.

### Reporting summary

Further information on research design is available in the Nature Portfolio Reporting Summary linked to this article.

## Data availability

All raw data generated in this paper have been deposited in the University of Cambridge Symplectic Elements database and can be accessed here: https://doi.org/10.17863/CAM.99108. Source data for each figure are provided as a Source Data file. Source data are provided with this paper.

## Code availability

Code is provided with this paper at https://doi.org/10.17863/CAM.99108.

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

## Acknowledgements

The authors would like to thank Friederike Hesse and Kevin Brindle from the University of Cambridge for supplying [2,3-$^2$H$_2$]-fumarate and Liam Wilson, Nicholas Tomkinson, William Tipping and Duncan Graham from the University of Strathclyde for supplying mitokyne. We are grateful to Marilena Oraiopoulou and Seema Chandana Bachoo for reproducing some of the results and for valuable discussions. We appreciate the valuable contributions of project students Rachel Xin Ran Lim and Oliver Powell. Dominique-Laurent Couturier is thanked for useful discussions. We would also like to thank the Light Microscopy Core Facility and the Histopathology Core Facility of the CRUK Cambridge Institute. This work was funded by CRUK (C47594/A16267, C14303/A17197, C9545/A29580) awards to SEB.

## Author contributions

M.K. provided intellectual input to the project, designed and performed the experiments, performed the statistical analysis and generated the figures. J.S., M.S.M. and T.Y. contributed to the formulation of the project. J.S. and K.C. performed initial experiments. J.S. and M.K. performed the partial least squares discriminant analysis. F.J. supported the Raman microscope and cell chamber. V.Z. and A.S. provided tissue sections. C.F. and S.E.B. supervised the experiments, conceived and oversaw the study. All authors read and approved the manuscript.

## Competing interests

The authors declare no competing interests.
