## [Peer Review File · Nature Communications]

Reviewers' Comments:

Reviewer #1:

Remarks to the Author:

In this study, Drs. Bohndiek and Frezza and their colleagues report that Raman spectroscopy (RS) can directly detect fumarate concentration in living cells *in vivo* and in animal tissues *ex vivo*. The concentration of fumarate can be used to distinguish between fumarate hydratase (Fh1)-deficient and Fh1-proficient cells. Furthermore, RS analysis reveals the spatial compartmentalization of fumarate within cellular organelles. Overall, this study opens new avenues for the study of fumarate. Some issues need to be addressed before it can be considered for publication in the journal.

1. In Fig. 1, I am curious as to why the highest Raman peak intensities were observed for fumarate compared to the other TCA cycle intermediates. Does this imply a limitation of the RS and may raise concerns about the reliability of this stage in metabolite determination?
2. Some of the experiments need more controls to solidify the results, for example in Fig. 2B it would be better to include a Fh1-wt group fed with ¹³C-L-glutamine.
3. Although the authors claim to have used two previously characterized mouse FH-deficient cell lines (Fh1^{-/-} Cl1 and Fh1^{-/-} Cl19) and their isogenic control (Fh1^{fl/fl}) to study RS detection of fumarate, validation of the efficacy of FH knockout in these cell lines is still required.
4. Similarly, in Fig. 4, we don't know whether or not FH was knocked out in these mouse tissues.
5. In Fig. 3D, Comparing the organelle distribution maps shown in the k-means cluster maps on the right, the fumarate concentration maps on the left show that fumarate is also detected outside the cell, is this noise or is it a real situation? Could the authors explain or comment on this?
6. What tissue was used in Fig. 4? No direct and clear information was given in the text and legends, it appears to be kidney from the following description.
7. Some of the labels and descriptions of the figures are confusing, e.g. FIGURE 4, according to the text, diagrams and LEGENDS, I can't tell what the difference is between FH^{-/-}, induced, KO?
8. Since 'LCMS is not well suited for dynamic studies in living cells or tissues', I don't think RS is well suited for dynamic studies in tissues either currently. Detection of fumarate in tissues by RS requires frozen tissue sections after euthanasia of mice, and the real-time dynamic level of fumarate in tissues cannot be analyzed. Moreover, for metabolites of different chemical nature, sample processing *in vitro* usually alters metabolite stability or modification status.
9. Overall, I think RS technology at this stage may be not as convenient as LC-MS and also has some limitations. However, RS technology may be a good choice as an auxiliary method for fumarate study together with LC-MS and fumarate chemical probes.

Reviewer #2:

Remarks to the Author:

This work demonstrates that Raman spectromicroscopy (RS) can directly detect and map the oncometabolite fumarate in living cells and frozen tissues, and shows that RS can discriminate Fh1-deficient and -proficient cells and tissues, while revealing the previously unseen spatial distribution of fumarate concentrations according to cellular compartment. These findings open a new avenue to metabolic imaging of small molecules.

However, the manuscript overstates about dynamic imaging, which is supposed to be the key advantage compared with traditional methods. Actually, there is no evidence provided to show this method can be used for dynamic imaging of fumarate. According to the methods section, RS with 785nm excitation takes ~120min to cover a single cell area and RS with 532nm excitation takes ~10min to cover a single cell area. Such slow speed is not feasible for dynamic imaging of metabolites. Thus, it does not meet the high standard of NC for publication.

Reviewer #3:

Remarks to the Author:

In this work, the authors show for the first time that Raman spectroscopy (RS) can directly detect fumarate in living cells *in vivo* and animal tissues *ex vivo*. Using the observed linear relationship between Raman scattered intensity and fumarate concentration, they demonstrate that RS can distinguish between Fumarate hydratase (Fh1)-deficient and Fh1-proficient cells based on their fumarate concentration. The authors suggest that RS could be adopted as a valuable tool for small molecule metabolic imaging, enabling *in situ* dynamic evaluation of fumarate compartmentalization.

This paper provides some interesting ideas for detecting fumarate in living cells *in vivo* and animal tissues *ex vivo*, while the novelty about using linear relationship between Raman scattered intensity and fumarate concentration seems not pretty sufficient to Nature Communications. Some results should be discussed more deeply. The following revisions are needed before its publication.

1. As we all know that normal Raman scattered intensity is pretty weak, it should be not easy to detect such weak scattered signal without any enhancement means, especially detecting fumarate in living cells *in vivo* and animal tissues *ex vivo*, where the complex background signals are not easy to be suppressed. The authors should provide more explanation about such issue.

2. Regarding as the Density functional theory (DFT) simulations performed to estimate the anticipated Raman cross sections of fumarate and comparable TCA cycle metabolites, generally the Raman vibrational mode positions and the corresponding intensities could be obtained, is there any calibrations about the anticipated Raman cross sections. Moreover, DFT calculations examining a range of fumarate and its ions are in gas-phase, but the following experimental results are in solid-state and solution phase, should solvation model or other solid-state DFT calculations be considered to provide more accurate description about comparing the simulations to experimental Raman spectra acquired from powders and aqueous solutions?

3. The authors mentioned that Fumarate displays prominent Raman spectral features compared to other TCA cycle intermediates shown in Figure 1A and B, actually it is not very clear why such specific Raman spectra could be assigned as various TCA cycle intermediates. Is there any time-resolved or temperature dependent metabolite components?

4. Some results should be discussed more deeply including the following conclusion with more useful and specific information about this work.

Response to reviewer comments

Our changes in the manuscript are denoted with blue text, as are our responses to the reviewer comments below.

Reviewer #1 (Remarks to the Author):

In this study, Drs. Bohndiek and Frezza and their colleagues report that Raman spectroscopy (RS) can directly detect fumarate concentration in living cells in vivo and in animal tissues ex vivo. The concentration of fumarate can be used to distinguish between fumarate hydratase (Fh1)-deficient and Fh1-proficient cells. Furthermore, RS analysis reveals the spatial compartmentalization of fumarate within cellular organelles. Overall, this study opens new avenues for the study of fumarate. Some issues need to be addressed before it can be considered for publication in the journal.

We thank the referee for their positive remarks on our manuscript and appreciate their acknowledgement of the potential of this study for opening new avenues opened for the study of fumarate biology.

1. In Fig. 1, I am curious as to why the highest Raman peak intensities were observed for fumarate compared to the other TCA cycle intermediates. Does this imply a limitation of the RS and may raise concerns about the reliability of this strategy in metabolite determination?

As the reviewer correctly notes, we observed the highest Raman peak intensities for fumarate, with lower intensities for other TCA cycle intermediates. Fumarate exhibited the highest peak intensities in density functional theory simulations, as well as in powder and aqueous spectra, indicating a higher Raman scattering cross section (related to polarizability) for fumarate than other metabolites.

Our aim in the current work is to report our first successful direct imaging and spatial mapping of a small molecule metabolite in living cells and tissues, hence we focused on fumarate given its high Raman scattering intensity and importance in the context of cellular metabolism and disease.

The lower Raman scattering cross section of other TCA intermediates is indeed a potential limitation and implies that it could be more challenging to obtain quantification and spatial mapping for other metabolites using spontaneous Raman spectroscopy. The interplay of metabolite and cellular vibrational modes and hence peak positions also need consideration, as we highlight in our manuscript.

Nonetheless, Coherent Raman techniques such as stimulated Raman spectroscopy and stimulated Raman photothermal microscopy can selectively stimulate a known vibrational mode using a pump-probe approach. By selectively exciting targeted vibrational modes that distinguish metabolites against other cellular components, we expect to be able to extend the application of Raman spectroscopy to other metabolites. Although coherent techniques require more specialist microscopy equipment, they also afford substantially greater sensitivity and imaging speed compared to conventional spontaneous Raman microscopy, offering a promising route to enable broader metabolite determination using Raman scattering.

We have now expanded our discussion to fully evaluate these points (lines 596-605 and 612-622).

2. Some of the experiments need more controls to solidify the results, for example in Fig. 2B it would be better to include a Fh1-wt group fed with ¹³C-L-glutamine.

We agree that this is an important control experiment. We have now performed this experiment and have added the associated data to the supplementary information (Figure S12) to retain clarity of the spectral shifts observed in the main text (Figure 2). The new figure is referenced in the main manuscript on lines 470-471.

3. Although the authors claim to have used two previously characterized mouse FH-deficient cell lines (Fh1^{-/-} Cl1 and Fh1^{-/-} Cl19) and their isogenic control (Fh1^{fl/fl}) to study RS detection of fumarate, validation of the efficacy of FH knockout in these cell lines is still required.

We apologise for not including further information in the manuscript. In the present study, the correspondence of fumarate concentration measurements derived from Raman spectroscopy with prior measurements acquired using LC-MS was taken to confirm the efficacy of the FH knockout.

PCR experiments could be performed for further confirmation; however, these would be on a later passage of the cell stocks as the original stocks used for the mapping experiments have unfortunately since been exhausted.

4. Similarly, in Fig. 4, we don't know whether or not FH was knocked out in these mouse tissues.

We apologise for the lack of clarity in the description of our methods. The tissues used for this study were prepared from mice bred by co-author Zecchini & Frezza during their recently published study, Zecchini, V. *et al.* Fumarate induces vesicular release of mtDNA to drive innate immunity. *Nature* **615**, 499–506 (2023). In the published work, extensive validation of Fh1 expression, fumarate intensity and the fumarate marker 2-succinocysteine (2SC) intensity was performed in kidney tissues harvested from the mice, as illustrated below (reproduced from Figure 1 of published paper). The Fh1 mRNA expression levels were measured by qRT-PCR and metabolite abundances were extracted from liquid chromatography-mass spectrometry (LC-MS).

In the present work, the Raman spectroscopy measurements were performed blinded on sections of these kidney tissues harvested at day 10 post-induction, when the maximum difference is expected between the Fh1^{fl/fl} and Fh1^{-/-} groups. Raman analysis was performed to assign the tissues to Fh1^{fl/fl} and Fh1^{-/-} groups and their experimental status was confirmed only after these assignments.

We have now updated our methods section (lines 218-220, 226-228, 286-289) accordingly.

5. In Fig. 3D, Comparing the organelle distribution maps shown in the k-means cluster maps on the right, the fumarate concentration maps on the left show that fumarate is also detected outside the cell, is this noise or is it a real situation? Could the authors explain or comment on this?

Indeed, this is an observation worthy of further explanation. The fumarate positivity observed outside of the cell can indeed be attributed to noise. It occurs because the same fitting algorithm is applied to all recorded spectra within the rectangular imaging region, without any mask applied to delineate the cell. There, the polynomial fitting procedure when applied to the (locally non-existent) CH₂ deformation peak can sometimes still converge to a fit, even on the background spectrum. A mask could be used to apply the fitting algorithm only to spectra within the cell region, however, we felt it was fairer to show the fitting to the full region to indicate more clearly the noise threshold.

We have now updated the results section (lines 548-552) to explain this as requested by the reviewer.

6. What tissue was used in Fig. 4? No direct and clear information was given in the text and legends, it appears to be kidney from the following description.

The tissue is indeed kidney, and this has now been clarified in the caption (line 581).

7. Some of the labels and descriptions of the figures are confusing, e.g. FIGURE 4, according to the text, diagrams and LEGENDS, I can't tell what the difference is between FH/-, induced, KO?

We thank the reviewer for their feedback that the text and caption contained confusing description with different naming conventions combined, due to need to distinguish tissues from the cellular materials. We have now made the text, figure and caption consistent (lines 580-586).

8. Since 'LCMS is not well suited for dynamic studies in living cells or tissues', I don't think RS is well suited for dynamic studies in tissues either currently. Detection of fumarate in tissues by RS requires frozen tissue sections after euthanasia of mice, and the real-time dynamic level of fumarate in tissues cannot be analyzed. Moreover, for metabolites of different chemical nature, sample processing *in vitro* usually alters metabolite stability or modification status.

We agree that, currently, our method is not fast enough for real-time dynamic studies. Our intention was not to claim that we are able to perform dynamic imaging *in vivo* in real-time, rather to highlight the application of the method to living cells, so our use of the word dynamic was misleading. Here, we show for the first time a method for recording changes in cellular fumarate distribution *in situ* in a non-destructive manner. In the future, the non-destructive nature of the measurement applied to living cells could be extended to perform longitudinal studies of the intracellular fumarate distribution, but as the reviewer points out, this is rather different than true dynamic studies. We have updated the introduction manuscript to reflect this.

9. Overall, I think RS technology at this stage may be not as convenient as LC-MS and also has some limitations. However, RS technology may be a good choice as an auxiliary method for fumarate study together with LC-MS and fumarate chemical probes.

LC-MS destroys the cell and provides an ensemble average of the metabolite distribution so cannot perform live *in-situ* imaging at present, even though other approaches such as mass spec imaging are coming of age. Still, even these technologies rarely reach subcellular resolution and remain destructive for the cells or tissues. Raman spectroscopy overcomes these particular limitations, but as the reviewer points out, has limitations of its own in terms of scattering probabilities and throughput. Thus, the two technologies provide distinct complementary information compensating for respective limitations and enabling different scientific questions to be asked, as highlighted in our introduction (lines 93-105).

Reviewer #2 (Remarks to the Author):

This work demonstrates that Raman spectromicroscopy (RS) can directly detect and map the oncometabolite fumarate in living cells and frozen tissues, and shows that RS can discriminate Fh1-deficient and -proficient cells and tissues, while revealing the previously unseen spatial distribution of fumarate concentrations according to cellular compartment. These findings open a new avenue to metabolic imaging of small molecules.

We thank the referee for their positive feedback, showing clear understanding of the goal of our study, which is to previously unseen spatial distribution of fumarate concentrations according to cellular compartment. We agree that this approach could open a new avenue for metabolic imaging of small molecules.

However, the manuscript overstates about dynamic imaging, which is supposed to be the key advantage compared with traditional methods. Actually, there is no evidence provided to show this method can be used for dynamic imaging of fumarate. According to the methods section, RS with 785nm excitation takes ~120min to cover a single cell area and RS with 532nm excitation takes ~10min to cover a single cell area. Such slow speed is not feasible for dynamic imaging of metabolites. Thus, it does meet the high standard of NC for publication.

We apologise that our use of the word 'dynamic' has caused confusion and mutual misunderstanding. Our intention was not to claim that we are now able to perform dynamic imaging in time, which would be impractical in the current form as the reviewer rightfully remarks. Here, we intend to show a pathway to recording changes in cellular intrafumarate distribution by *in situ* imaging of cells that were plated at the same time.

Nonetheless, we do see potential for future dynamic imaging, as our current method could be easily combined with Coherent Raman techniques such as stimulated Raman spectroscopy and stimulated Raman photothermal microscopy, as noted in our response to reviewer 1.

We have now changed the word 'dynamic' to 'non-destructive' in the introduction. We have instead indicated the potential for future longitudinal imaging studies in the discussion section, significantly expanding the discussion of how this could be achieved (lines 602-630, 640-643).

We therefore believe that given the potential to open a new avenue for metabolic imaging this work merits consideration for Nature Comms.

Reviewer #3 (Remarks to the Author):

In this work, the authors show for the first time that Raman spectroscopy (RS) can directly detect fumarate in living cells *in vivo* and animal tissues *ex vivo*. Using the observed linear relationship between Raman scattered intensity and fumarate concentration, they demonstrate that RS can distinguish between Fumarate hydratase (Fh1)-deficient and Fh1-proficient cells based on their fumarate concentration. The authors suggest that RS could be adopted as a valuable tool for small molecule metabolic imaging, enabling *in situ* dynamic evaluation of fumarate compartmentalization. This paper provides some interesting ideas for detecting fumarate in living cells *in vivo* and animal tissues *ex vivo*, while the novelty about using linear relationship between Raman scattered intensity and fumarate concentration seems not pretty sufficient to Nature Communications. Some results should be discussed more deeply. The following revisions are needed before its publication.

We appreciate the Referee's comments on our work. We respectfully disagree that the novelty of the paper can be summarized by the use of the linear relationship between Raman scattered intensity and fumarate concentration. The novelty lies in the fact that metabolites, due to their small cross section, have not previously been able to be identified *in situ*. The only successful methods used previously, such as LC-MS, were able to detect average cellular concentrations by disrupting the cells to extract the metabolites, but cannot reveal *in situ* concentration distributions.

By combining a metabolite's vibrational properties in the context of fumarate hydratase deficiency, we show for the first time that it is possible to 'map' fumarate distributions inside living cells. Observing the distribution of a metabolite over the cell rather than extracting its average concentration is crucial to be able to understand cellular dynamics. Moreover, we demonstrate that the fumarate concentration is highest in the mitochondria, taking advantage of this newfound capability.

1. As we all know that normal Raman scattered intensity is pretty weak, it should be not easy to detect such weak scattered signal without any enhancement means, especially detecting fumarate in living cells *in vivo* and animal tissues *ex vivo*, where the complex background signals are not easy to be suppressed. The authors should provide more explanation about such issue.

The Referee is correct that Raman scattering is weak and not easy to detect. Obtaining robust Raman spectroscopy data requires carefully planned and well executed experiments, as is the case for many light microscopy tools. Nonetheless, that does not prohibit application in living cells *in vivo* or animal tissues *ex vivo*, as highlighted in the range of papers we reference in the Introduction, which have successfully applied commercial Raman instruments as well as home-built systems for both purposes.

As we note, the weak Raman scattering cross section limits our current demonstration using spontaneous Raman spectroscopy to fumarate and spatial distributions can only be confidently resolved in KO cells since they display elevated levels of fumarate. To achieve these results, we required relatively long integration times of 0.3s and a short excitation wavelength of 532 nm, but we undertook comprehensive studies to ensure these measurements were robust and to validate the findings using a range of other methods (at 785nm, using carbon-13 labelled substrates). We extensively discuss these trade-offs made to be able to detect fumarate in our Methods section.

Nonetheless, our current method could be easily combined with Coherent Raman techniques such as stimulated Raman spectroscopy and stimulated Raman photothermal microscopy, as noted in our

response to reviewer 1. We have now substantially expanded our discussion to more directly address these challenges and future perspectives on how they could be addressed (lines 602-630, 640-643).

2. Regarding as the Density functional theory (DFT) simulations performed to estimate the anticipated Raman cross sections of fumarate and comparable TCA cycle metabolites, generally the Raman vibrational mode positions and the corresponding intensities could be obtained, is there any calibrations about the anticipated Raman cross sections. Moreover, DFT calculations examining a range of fumarate and its ions are in gas-phase, but the following experimental results are in solid-state and solution phase, should solvation model or other solid-state DFT calculations be considered to provide more accurate description about comparing the simulations to experimental Raman spectra acquired from powders and aqueous solutions?

In this case, the DFT calculations serve as a means of calibration, showing that the cross section of fumarate is higher than that of other metabolites, although an absolute measure for the cross section cannot be found in this way. Additionally, the DFT simulations, together with the experimental data, were used to confirm that there is no overlap of the main vibrational modes of fumarate with those of other metabolites and to identify how the peaks relate to the other vibrational modes recorded with cellular spectra more generally.

The Referee is correct regarding solvation and we thank them for identifying this as it was reported incorrectly. A solvation model was applied in the form of a self-consistent reaction field, more specifically the polarizable continuum model for water. This is now reported correctly in the methods (lines 136-137), and we thank the reviewer for pointing this out.

3. The authors mentioned that Fumarate displays prominent Raman spectral features compared to other TCA cycle intermediates shown in Figure 1A and B, actually it is not very clear why such specific Raman spectra could be assigned as various TCA cycle intermediates. Is there any time-resolved or temperature dependent metabolite components?

Raman features are not expected to be time- or temperature-dependent, since fumarate does not undergo any conformational changes over time or with temperature. The reviewer is correct that metabolites of fumarate would display different vibrational modes, however, fumarate hydratase converts fumarate to malate, which we have shown via measurements and DFT simulations has a smaller cross-section than fumarate at unfavourable vibrational positions relative to the cellular spectra. A fumarate adduct that may be formed is 2-succinocysteine, for which we found experimentally that the spectrum lacks salient features. We also saw only the expected changes in the fumarate peaks when incubating with $^{13}\text{C}_5$ -labelled glutamine, which gives rise to $^{13}\text{C}_4$; if metabolic products were interfering with the spectrum, we would have seen additional peak shifts arising from ^{13}C labelling.

Taken together, we do not expect strong signals from metabolites or complexes of fumarate to interfere with the measurements obtained.

4. Some results should be discussed more deeply including the following conclusion with more useful and specific information about this work.

We have now expanded our discussion section to more comprehensively evaluate our findings and clarified the potential of coherent Raman spectroscopy tools to overcome sensitivity limitations.

Reviewers' Comments:

Reviewer #1:

Remarks to the Author:

I appreciate the authors' efforts to address my concerns. I think they have addressed most of my concerns, although I still think it would be preferable to show validation of FH expression in tissues from Fh1f/fl and Fh1-/- mice, but learning that these mice came from Dr. Frezza's lab, I was convinced. The paper can now be considered for acceptance, I think.

Reviewer #2:

Remarks to the Author:

The authors have addressed my concerns.

Reviewer #3:

Remarks to the Author:

The authors provided replies to some critical issues raised in the previous review stage together with English editing on entire manuscript. This manuscript would be acceptable, unless otherwise decided by other reviewers.